# Bio-activity prediction of drug candidate compounds targeting SARS-Cov-2 using machine learning approaches

**Faisal Bin Ashraf[1,2], Sanjida Akter[3], Sumona Hoque Mumu[4], Muhammad Usama Islam[5], Jasim Uddin**[6]*

**1** Department of Computer Science and Engineering, Brac University, Dhaka, Bangladesh, **2** Department of Computer Science and Engineering, University of California, Riverside, California, United States of America, **3** Department of Cell Molecular and Developmental Biology, University of California, Riverside, California, United States of America, **4** School of Kinesiology, University of Louisiana at Lafayette, Lafayette, Louisiana, United States of America, **5** School of Computing and Informatics, University of Louisiana at Lafayette, Lafayette, Louisiana, United States of America, **6** Department of Applied Computing and Engineering, Cardiff School of Technologies, Cardiff Metropolitan University, Cardiff, Wales, United Kingdom

* juddin@cardiffmet.ac.uk

**Data Availability Statement:** All relevant data, data extraction, and coding are shared at the link: (https://github.com/fbabd/bioactivity-against-SARS-CoV-2).

## Abstract

The SARS-CoV-2 3CLpro protein is one of the key therapeutic targets of interest for COVID-19 due to its critical role in viral replication, various high-quality protein crystal structures, and as a basis for computationally screening for compounds with improved inhibitory activity, bio-availability, and ADMETox properties. The ChEMBL and PubChem database contains experimental data from screening small molecules against SARS-CoV-2 3CLpro, which expands the opportunity to learn the pattern and design a computational model that can predict the potency of any drug compound against coronavirus before in-vitro and in-vivo testing. In this study, Utilizing several descriptors, we evaluated 27 machine learning classifiers. We also developed a neural network model that can correctly identify bioactive and inactive chemicals with 91% accuracy, on CheMBL data and 93% accuracy on combined data on both CheMBL and Pubchem. The F1-score for inactive and active compounds was 93% and 94%, respectively. SHAP (SHapley Additive exPlanations) on XGB classifier to find important fingerprints from the PaDEL descriptors for this task. The results indicated that the PaDEL descriptors were effective in predicting bioactivity, the proposed neural network design was efficient, and the Explanatory factor through SHAP correctly identified the important fingertips. In addition, we validated the effectiveness of our proposed model using a large dataset encompassing over 100,000 molecules. This research employed various molecular descriptors to discover the optimal one for this task. To evaluate the effectiveness of these possible medications against SARS-CoV-2, more in-vitro and in-vivo research is required.

## 1 Introduction

Recent epidemic outbreaks have emphasized the importance of establishing affordable cost treatments. Discovering new tiny molecules known as ligands along with substantial bioactive

**Funding:** The authors received no specific funding for this work.

**Competing interests:** NO - The authors have declared that no competing interests exist.

components against target proteins, also known as receptors, is an important step in early drug design. A substance's bioactivity, which reflects its potency and ability to have a biological effect, is critical to its pharmacological effects.

Classically, promising compounds are screened using low or high throughput experimental bioassays; however, these approaches are costly and time-consuming, rendering them unsustainable for large molecules like proteins. Computational approaches have made significant strides in accurately and efficiently predicting the biological activity of both small and large molecules to overcome these challenges. This has resulted in the development of competitive inhibitors, which are considered to be bioactive small molecules with a specific binding affinity but can also be subsequently experimentally evaluated.

The COVID-19 pandemic has increased the demand for new antiviral medications or therapies. One of the most significant challenges is the time required to finalize the chemicals for vaccine formulation, which can stymie vaccine development and have serious consequences. Although several trials for many pharmaceutical companies have been successful, using artificial intelligence to predict potential chemicals for vaccine formulation could significantly speed up the process and save lives. Studies have been undertaken in this arena to employ the neural network for vaccine formulation. However, the majority of these face accuracy challenges [1–3]. Traditional HTS campaigns are often limited to 1–2 million compounds owing to the high costs and operational bottlenecks that limit the chance for lead identification [1, 2]. High costs and low hit rates limit the identification of anti-SARS-CoV-2 compounds through traditional high-throughput screening (HTS) assays which is pointed out in Xu's work [3]. The Virtual screening methods like QSAR [4] depend on the availability of chemical structure information to infer predictions, limiting the power to discover new chemical scaffolds and applicability of such methods to querying only in the close structural vicinity of already known ligand structures and drug targets [3–5]. Consequently, biological activity predictions made on chemicals with structure types not included in the training set are often unreliable, rendering to applicability domain (AD) issue [4, 5]. The extent of application of such a cross-validated, predictive model to discovering fundamental new drugs is still hypothetical. The lack of clarity in situational benefit or hindrance in transfer and multitask learning and uncertainty about the variables that govern how semi-supervised learning affects model prediction performance exists [5]. Additional limitations relate to the distribution of data and the type of data used in drug discovery as pointed out in opinion voiced out in Baskin's work [6]. Several research work has shown promise of utilizing machine learning and neural networks for bioinformatics research, including drug discovery [7] where, as some others voiced concerns over representation of data across the chemical space accurately [6]. Substantial challenges on data interpretation was pointed out in [8] as well as some implications on data and phenomena understanding became crucial factors which are discussed in Polishchuk and colleagues work [9].

The bulk of vaccine composition in research and development is typically undertaken in clinical setting, while alternative medication formulations are evaluated to discover the best optimal option. This is an ideal opportunity to consider implementing artificial intelligence (AI) to produce more rapid and efficient outcomes. AI algorithms can analyze massive amounts of data and predict the efficacy of various drug combinations, reducing the time and resources required for clinical trials and increasing the likelihood of developing a successful vaccine.

In this study we investigated a variety of ensembles and classical machine learning techniques to determine the biocompatibility of compounds against the SARS-CoV-2 3CLpro protein. It has also been explored the impact of several molecule representations on prediction. The study presented an efficient neural network-based design and utilized it to identify a few promising molecules.

We obtained 93% accuracy in our result and a f-1 score of 0.94 through experimentation, which indicates that the proposed neural network design is efficient, after tweaking through more than 25 traditional and ensemble classifiers with two descriptors, namely PaDEL and Lipinski. Furthermore, we have also reported important fingerprints with their bit position, substructures as well as effects followed by Shapley values based explanation to imrprove the interpretability and explainability of our predictions. To verify the findings, we also looked at several molecular description methods in order to determine which one performed better in the classification activity and compared the findings to the standard ones.

The following is a breakdown of the paper's structure. The application of machine learning techniques in predicting bioactivity is discussed in Section 2 and the methods used for data collection, curation, and implementation are described in Section 3. All of the experimental results that are addressed in Section 5 are shown in Section 4. By using our approach to compare a collection of compounds against the SARS coronavirus, Section 6 finds additional candidate compounds, and Section 7 closes the paper.

## 2 Machine learning in predicting bioactivity

Machine learning techniques have emerged as powerful tools for predicting the bioactivity of existing drugs based on their molecular structure and properties. This methodology facilitates the identification of novel drug candidates, repurposing of existing drugs, and optimization of drug design. Different molecular representations, such as fingerprints, descriptors, graphs, and SMILES strings, capture the structural features essential for analysis. These features are then utilized as inputs for diverse machine learning models, including deep neural networks, support vector machines, random forests, and decision trees, which effectively learn the intricate relationship between drug structure and activity. Several studies have reported significant findings in this area, using machine learning techniques to predict the bioactivity of existing drugs. Mongia et al. [10] developed an interpretable machine learning approach to identify novel antibiotics with diverse mechanisms of action, leading to the discovery of bioactive molecules with potential antibacterial activity and novel binding modes. These studies highlight the efficacy of machine learning in predicting bioactivity and its potential in drug discovery.

In recent years, machine learning (ML) approaches have been effectively employed to forecast the biological actions of substances. In order to demonstrate the promise and efficacy of these methods in the early stages of drug discovery, Lane's team [11] conducted a thorough evaluation of various ML algorithm and Santana and his colleagues [12] on 5000 datasets from ChEMBL. With the help of bioactivity information on human Carbonic anhydrase (hCA II, hCA IX, and hCA XII) found on ChEMBL, a number of machine learning classifiers were developed [13]. In this experiment, each molecule was represented by one of 92 molecular descriptors, and the Extra Tree classifier was determined to be the most effective. To decrease misclassification, a likelihood score for each class was computed and applied. Baassi and colleagues discusses the in-silico design of a potential new HIV-1 protease inhibitor [14] where the authors reiterated the use of in-silico methods for drug design, which can potentially speed up the drug development process.

For quantitative activity prediction, predictions were made using deep learning techniques like Graph CNN [15]. Only two-dimensional structural feature data from 127 target proteins were used in this investigation. For the ChEMBL datasets, GCN outperformed CNN, RF, and FNN in terms of performance. Preprocessing is a crucial step in ML implementation, and this study followed prior studies by using the maximum value when many values were discovered for the same compound-target pairings [16, 17].

A neural network imputation method that can learn from incomplete bioactivity data has been proposed. The correlation between molecular descriptors and bioactivity, as well as between different bioassays, was used by the authors [16]. This model can also calculate the confidence of the prediction. This is a better model for difficult datasets where the compounds are poorly represented. Galushka and colleagues [18] presented a work inspired by variational autoencoder in which they predicted the fingerprint of bioactive compounds. Using 1024 latent vectors and ChEMBL data, the model was able to regenerate 90% of the compounds' SMILES. Although such errors in regeneration are not accepted, their contribution to locating the SMILE representation's latent space can increase efficiency in classification and regression tasks. CSConv2d, which has a convolutional block attention module (CBAM), was proposed for drug-target interaction (DTI) prediction [19]. A deep learning framework that requires no extra computation was proposed to compute a valid and efficient confidence interval [20]. It has the potential to expand the use of deep learning in early-stage drug discovery with reliability.

Machine Learning technologies have significant opportunities and potential to combat COVID-19 through their use in predicting compounds for drugs and vaccines [21, 22]. It is possible to find the required drug using a drug and an open chemical database as inputs. Jha and team [23] proposed a deep learning approach based on Logistic Regression, SVM, and Random Forest after QSAR modeling. Deep learning was used to learn from the OPLRAreg algorithm's molecular descriptors. Another study conducted by [24] compared two generative models—JT-VAE and DQN—in order to identify small candidate molecules against SARS-CoV-2. They discovered that DQN performed better in terms of score, and that JT-VAE produced molecules that were structurally similar to those in the database. Potential candidates were identified in another study by [25] via RDOCK virtual screening of the ChEMBL dataset, and potential drugs were listed in their paper. Another paper by [26] used Random Forest combined with Recursive Feature Elimination and Cross-Validation to build SVM classifiers. Using this method, they achieved an accuracy of 88% on PostEra COVID-19 Moonshoot public activity data. Santana and his colleagues [12] used ULMFit to train chemical models and design a classifier for bioactivity prediction. For the classification task, they used transfer learning and obtained more than 90% valid, novel, and diverse candidates. In the work by [27], QSAR and molecular docking were used to identify novel inhibitors against SARS-CoV-2 3CLpro. They proposed five candidate components for further in-vitro and in-vivo coronavirus research. Another study to find SARS-CoV-2 3CLpro inhibitors by [28] combined virtual screening, molecular dynamics, machine learning, and in vitro analysis.

## 3 Materials and methods

### 3.1 Data collection and curation

We have collected data for bioactivity prediction from two different database. First one is ChEMBL by [29] database where a number of experimental results are stored against SARS-CoV-2. The second one is BioAssay data from PubChem database [30, 31] that contains around 300,000 compounds activity against SARS-Cov-2 3CLpro.

Experimental data from ChEMBL targeting coronavirus (https://www.ebi.ac.uk/chembl/g/ #search_results/targets/query=coronavirus), single protein was used in this study. The number of bioactive and inactive compounds used in these trials is displayed in Table 1. The activities were measured using a variety of standards, principally inhibition (%), IC50 (nM), and Ki (nM). Which molecules can be treated with lower dosages depends on the half maximum inhibitory concentration (IC50), which measures the amount of medication required to inhibit a target by 50%. Therefore, we have used the estimated data of the IC50 standard.

**Table 1. Bioactivity datasets from ChEMBL.**

| ChEMBL Dataset | Active Compound | Inactive Compound | Total Compounds |
|---|---|---|---|
| CHEMBL3927 | 15 | 104 | 119 |
| CHEMBL4523582 | 5 | 156 | 161 |
| CHEMBL5118 | 79 | 34 | 113 |

It is essential to keep in mind that the IC50 values reported in various research can change according on the experimental setup. For instance, several studies have documented the IC50 values for the FDA-approved medication remdesivir, which is effective against SARS-CoV-2. Remdesivir had an IC50 of 0.77 $\mu$M in Vero E6 cells infected with SARS-CoV-2 [32]. Remdesivir's IC50 was found to be 0.16 $\mu$M in another investigation [33] in human airway epithelial cells infected with SARS-CoV-2. Therefore, the values for this candidate is in the range of 0.16–0.77 $\mu$M Based on the IC50 values against SARS-CoV-3CLpro protein, we categorized the data in our curated dataset as active, intermediate, or inactive using the following criteria —Active : IC50 value $< 1000 nM$, Intermediate : $1000 \leq$ IC50 value $< 10,000 nM$, Inactive : IC50 value $> 10,000$ nM. Since powerful inhibitors had IC50 values less than 1000 nM, these threshold values were chosen since models created using them outperformed those discovered through [34]. Additionally, the activity levels as inhibitors are no longer assessed by [35] if the IC50 value hits 1000 nM. As a result, we classified the data as Intermediate and Inactive as the value climbed. If the IC50 value is lower, then less of the chemical is needed to cause inhibition. The intermediate active substances were excluded from our analysis. We used information from both active and inactive chemicals for our models.

We then searched the Pubchem database for additional experimental data and discovered a few tests against SARS-CoV-2 3clpro. For machine learning, we gathered all the datasets associated with the works by [30, 31] and combined them. We have collected approximately 300,000 molecular bioactivity records from this data repository. However, the number of active molecules is significantly lower than the number of inactive molecules, causing the dataset to be unbalanced. The specifics of these gathered datasets are displayed in Table 2.

**Table 2. Bioactivity datasets from Pubchem.**

| PubChem AID | Active Compound | Inactive Compound | Total Tested Compounds |
|---|---|---|---|
| AID1706 | 405 | 290,321 | 290,726 |
| AID1879 | 136 | 244 | 380 |
| AID1890 | 44 | 57 | 101 |
| AID1944 | 19 | 82 | 101 |
| AID435015 | 0 | 1 | 1 |
| AID488877 | 0 | 1 | 1 |
| AID488958 | 9 | 5 | 14 |
| AID488967 | 15 | 17 | 32 |
| AID488984 | 10 | 93 | 103 |
| AID488999 | 3 | 1 | 4 |
| AID493245 | 3 | 3 | 6 |
| AID588771 | 5 | 5 | 10 |
| AID588772 | 14 | 14 | 28 |
| AID588786 | 3 | 7 | 10 |
| AID602486 | 0 | 1 | 1 |
| AID602487 | 0 | 5 | 5 |

### 3.2 Molecular descriptors calculation

The 3D chemical structures must be converted into a mathematical form that the computer can understand. Molecular descriptors are traits of molecules that are determined by an algorithm. In order to predict biological activity, chemical substances are given molecular descriptors. These descriptors are then used to build a quantitative structure-activity relationship (QSAR) model. The descriptor values are employed for a range of tasks, such as drug design and similarity searches, and they indicate the physical and chemical characteristics of the molecule. In this work, two descriptors—"Lipinski" proposed by [36, 37]—have been used. Fig 1 depicts the study's workflow.

It is crucial to carefully evaluate the selection of descriptor in order to create a representation that works for a deep learning framework. The physical and chemical characteristics of a molecule, such as its solubility, lipophilicity, and molecular weight, are described by physiochemical descriptors. For the design and improvement of drugs, these characteristics may be crucial. On the other hand, molecular descriptors provide a more thorough description of a molecule's structure by taking into account its atom connections and molecular geometry. It is crucial to thoroughly consider the advantages and disadvantages of each approach before selecting the one that is best suited for what we are trying to accomplish.

Among the software tools commonly used for molecular description and computational chemistry, PaDEL stands out as an excellent choice for high-throughput QSAR modeling. PaDEL's focus on efficient generation of a wide range of molecular descriptors, including 1D,

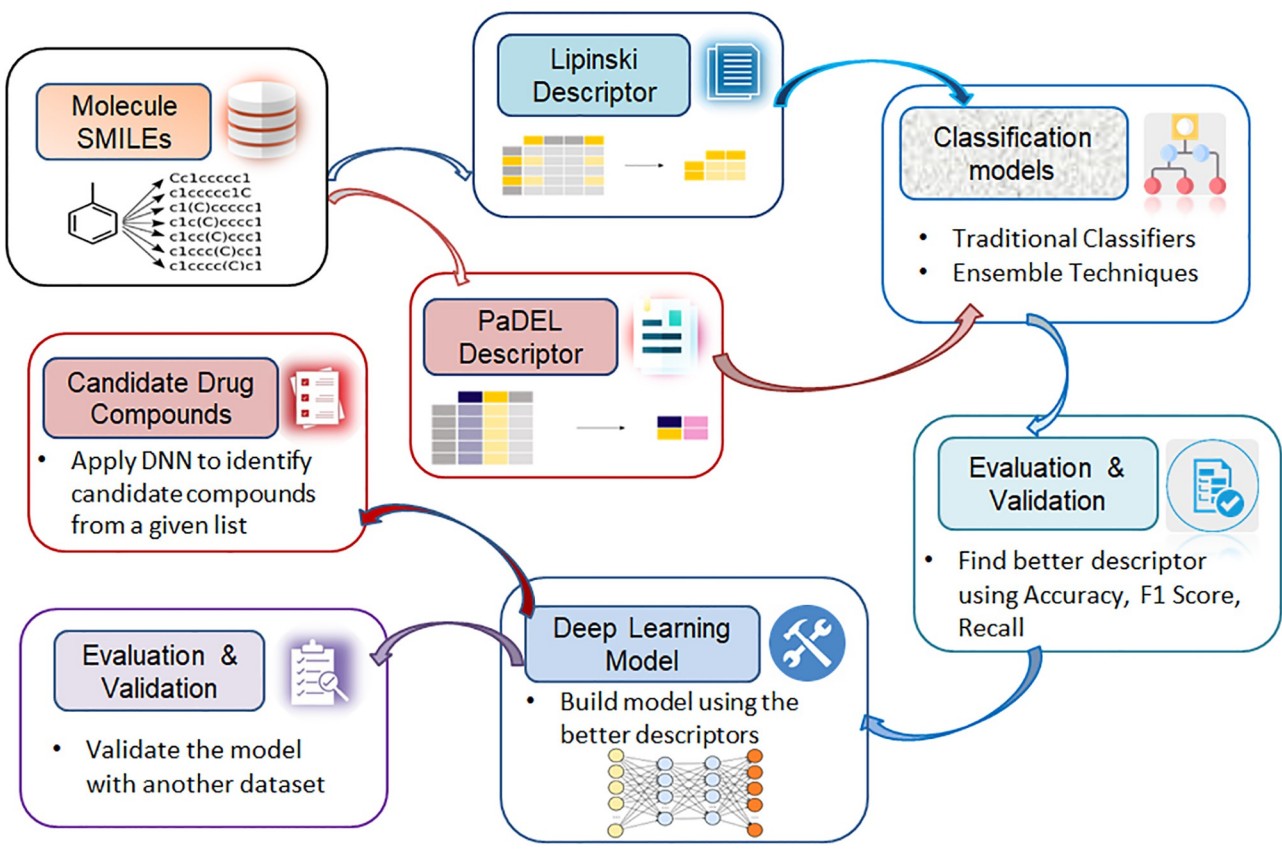

**Fig 1. Workflow of the study.**

2D, and 3D descriptors, makes it particularly advantageous for our experimentation. Its ability to swiftly calculate descriptors for large chemical databases enables rapid screening and analysis, while its user-friendly interface ensures accessibility for researchers of varying expertise. With PaDEL, we can efficiently explore the intricate relationships between molecular structures and activities, advancing our understanding of drug design and discovery.

**3.2.1 Lipinski descriptors.** Lipinski's general rule of thumb describes the drugability of a specific molecule. It is based on pharmacological properties known as pharmacokinetics and aids in determining whether a molecule has the necessary chemical and physical properties to be orally accessible. The conditions are listed here, and if two or more are broken, the molecule is unlikely to be consumed.

- Molecular mass not greater than 500 Dalton

- Octanol-water partition coefficient (LogP) not greater than 5

- Hydrogen bond donors not greater than 5

- Hydrogen bond acceptors not greater than 10

**3.2.2 PaDEL descriptors.** PaDEL is a program that calculates molecular descriptors and fingerprints developed by [37]. The chemical development kit is used to generate ten different types of fingerprints, 663 one- and two-dimensional descriptors, 134 three-dimensional descriptors, and other descriptors. This free and open-source software generates less descriptors and fingerprints such as atom type electrotopological state, molecule linear free energy relation, ring counts, and chemical substructure count.

## 3.3 Classification models for QSAR

From curated and preprocessed datasets, QSAR models were used to predict the bioactivity of substances using ensemble classifiers, conventional machine learning, and traditional machine learning, as well as neural network-based classifiers. We employed 26 conventional machine learning and ensemble classifiers, as well as the Scikit-learn [38], to semi-automate the machine learning activities. The classifiers were utilized using their default settings. However, these classifiers' performance might be enhanced by adjusting their hyperparameters. We were interested in learning which classifiers work best with their default settings.

The majority of traditional classification models are linear, identifying a linear relationship between the predicted and independent features. A linear relationship between a dependent variable $y$ and independent variables $x_1, x_2, \ldots, x_n$ can be represented by the following formula (Eq 1), where $w_1, w_2, \ldots, w_n$ are feature coefficients and $w_0$ is the line's intercept.

$$y = w_0 + w_1 * x_1 + w_2 * x_2 + \ldots + w_n * x_n \tag{1}$$

In this investigation, we employed linear logistic regression, Ridge, SGD, and passive-aggressive classifiers.

Support vector machines (SVMs) can be used for regression, classification, and outlier detection. When there are fewer samples than features in the dataset, these models perform well in higher-dimensional domains. We employed SVC, LinearSVC, and NuSVC classifiers in our study.

In addition to applying the Bayes theorem, naive Bayes classifiers make the incorrect assumption that each pair of characteristics is conditionally independent. We used Gaussian NB and Bernoulli NB in our investigation. The K-Neighbors Classifier and Nearest Centroid neighbor-based classifiers compute the class of the points using majority voting techniques.

Other classifiers were also applied on our dataset, including Decision Trees, Label Spreading, Label Propagation, and Dummy Classifiers.

By aggregating the predictions of several base estimators, ensemble techniques compute the final prediction. The predictions of various independent estimators are averaged in some ensemble approaches. By reducing the bias of the combined estimator, other ensemble approaches perform better. In this study, we employed a range of ensemble classifiers, including Random Forest, Bagging, Extra Tree, LGBM, XGB, and AdaBoost.

Finally, we developed a deep learning-based categorization model with plenty of dense layers. Neural networks provide a variety of benefits over traditional categorization techniques because of their ability to carefully evaluate and calculate all the features. Researchers can use hyperparameter tweaking to fine-tune model performance for the best outcomes. This is a fundamental aspect of machine learning, and selecting optimal hyperparameter values is critical for success. We investigated and experimented with many parameters, including the number of layers, the dropout ratio, and the number of nodes in each layer, and eventually developed an optimum design. Our proposed neural network for bioactivity prediction utilizing PaDEL features consists of eight hidden layers. In Table 3, the model's layers and their shapes are shown. Around 15M trainable parameters are available for this model. To address the over-fitting issue and improve the model, we added the dropout layers, which greatly improved performance. When creating the model, we modified the Adam optimizer to enhance the weights. The formula to update the weights by adding the Adam optimizer is shown in Eqs 2 and 3.

$$w_{t+1} = w_t - \alpha m_t \tag{2}$$

$$m_t = \beta m_{t-1} + (1 - \beta) \left[ \frac{\delta L}{\delta w_t} \right] \tag{3}$$

Here,
$m_i$ : Aggregate of gradients at time i
$w_i$ : Weights at time i
$\alpha$ : Learning rate
$\beta$ : Moving average

**Table 3. Neural network model architecture.**

| Layer (type) | Output Shape | Param # |
|---|:---:|:---:|
| input_1 (InputLayer) | (None, 881) | 0 |
| dense (Dense) | (None, 2048) | 1806336 |
| dropout (Dropout) | (None, 2048) | 0 |
| dense_1 (Dense) | (None, 4096) | 8392704 |
| dropout_1 (Dropout) | (None, 4096) | 0 |
| dense_2 (Dense) | (None, 1024) | 4195328 |
| dense_3 (Dense) | (None, 512) | 524800 |
| dense_4 (Dense) | (None, 256) | 131328 |
| dense_5 (Dense) | (None, 128) | 32896 |
| dense_6 (Dense) | (None, 2) | 258 |

=======================================

**Total params:** 15,083,650

**Trainable params:** 15,083,650

**Non-trainable params:** 0

$\delta L$ : Derivative of Loss function

$\delta w_t$ : Derivative of weights at time t

## 4 Results

In this study, we looked at how well ensemble classifiers and conventional machine learning performed in predicting the bioactivity of several drugs that target coronavirus. To find the descriptor that is most appropriate for the task, we developed two different molecular descriptors for this experiment: Lipinski and PaDEL. There were two phases to the experiment. First, we used 26 conventional and ensemble classifiers independently to each of the two descriptors. From this experiments we will understand which descriptors are more suited for this task. After that, we created and put to the test a neural network architecture for this classification job using the more suitable chemical descriptors.

### 4.1 Lipinski Vs PaDEL descriptor

The results of traditional and ensemble machine learning classifiers on ChEMBL Dataset are displayed in Table 4. We calculated the accuracy, ROC-AUC, and F1 score for each classifier using both types of descriptors. SVC is found to be the most accurate classifier for Lipinski

**Table 4. Performance of traditional and ensemble classifiers on dataset collected from ChEMBL.**

| Model | PaDEL Desc. | | | Lipinski Desc. | | |
|---|---|---|---|---|---|---|
| | Acc. | AUC | F1 | Acc. | AUC | F1 |
| LogisticRegression | **0.87** | **0.87** | **0.87** | 0.65 | 0.65 | 0.65 |
| RidgeClassifier | 0.8 | 0.8 | 0.8 | 0.63 | 0.63 | 0.63 |
| RidgeClassifierCV | 0.78 | 0.78 | 0.78 | 0.63 | 0.63 | 0.63 |
| SGDClassifier | 0.8 | 0.8 | 0.8 | 0.44 | 0.42 | 0.39 |
| PassiveAggressiveClassifier | 0.83 | 0.83 | 0.83 | 0.68 | 0.67 | 0.68 |
| SVC | 0.85 | 0.85 | 0.85 | **0.76** | **0.76** | **0.76** |
| LinearSVC | 0.78 | 0.78 | 0.78 | 0.65 | 0.65 | 0.65 |
| NuSVC | **0.87** | **0.87** | **0.87** | 0.69 | 0.7 | 0.69 |
| GaussianNB | 0.76 | 0.76 | 0.76 | 0.74 | 0.73 | 0.74 |
| BernoulliNB | 0.57 | 0.57 | 0.56 | 0.61 | 0.62 | 0.61 |
| KNeighborsClassifier | 0.7 | 0.7 | 0.69 | 0.66 | 0.66 | 0.66 |
| NearestCentroid | 0.57 | 0.57 | 0.56 | 0.68 | 0.68 | 0.67 |
| DecisionTreeClassifier | 0.83 | 0.83 | 0.83 | 0.66 | 0.66 | 0.66 |
| LabelSpreading | 0.52 | 0.52 | 0.41 | 0.73 | 0.72 | 0.72 |
| LabelPropagation | 0.52 | 0.52 | 0.41 | 0.71 | 0.7 | 0.71 |
| DummyClassifier | 0.46 | 0.46 | 0.45 | 0.34 | 0.33 | 0.33 |
| CalibratedClassifierCV | 0.8 | 0.8 | 0.8 | 0.63 | 0.63 | 0.63 |
| LinearDiscriminantAnalysis | 0.7 | 0.7 | 0.7 | 0.65 | 0.65 | 0.65 |
| QuadraticDiscriminantAnalysis | 0.7 | 0.7 | 0.7 | **0.76** | **0.75** | **0.76** |
| RandomForestClassifier | 0.8 | 0.8 | 0.8 | 0.73 | 0.72 | 0.73 |
| BaggingClassifier | 0.8 | 0.8 | 0.8 | 0.71 | 0.7 | 0.71 |
| ExtraTreeClassifier | 0.76 | 0.76 | 0.76 | 0.71 | 0.71 | 0.71 |
| ExtraTreesClassifier | 0.8 | 0.8 | 0.8 | 0.74 | 0.74 | 0.74 |
| LGBMClassifier | 0.8 | 0.8 | 0.8 | 0.71 | 0.71 | 0.71 |
| XGBClassifier | 0.8 | 0.8 | 0.8 | 0.69 | 0.69 | 0.69 |
| AdaBoostClassifier | 0.8 | 0.8 | 0.8 | 0.61 | 0.61 | 0.61 |
| Perceptron | 0.76 | 0.76 | 0.76 | 0.71 | 0.71 | 0.71 |

descriptors, with an accuracy rate of 76%. While ensemble techniques had an accuracy rate of roughly 70%, linear classifiers did not perform effectively. However, when using PaDEL descriptors, we found that the NuSVC model had an accuracy of 87%. With PaDEL descriptors, nearly all classifiers performed better than Lipinski. About 80% of ensemble techniques were accurate. In both cases, support vector-based classifiers perform best when employed with their default parameters.

We discovered that PaDEL descriptors, as opposed to Lipinski, are more suitable for classification from the evaluation metrics. Since PaDEL generates many more molecular features than Lipinski, it does the categorization task better.

## 4.2 Deep learning model performance

The present study describes the development of a neural network based on PaDEL descriptors. The architecture of the implemented model is depicted in Table 3. The TensorFlow framework was employed for the implementation and conduct of trials. Initial training and assessment of the model was performed using a proprietary dataset for 200 iterations. The presence of a dropout layer prevented overfitting during the training phase. Furthermore, the validation accuracy of the model, with a maximum accuracy of 91%, indicates its efficacy on test data.

The performance of the proposed neural network model on the ChEMBL Dataset was evaluated and the results are presented in Table 5. The F1-score for inactive and active compounds was 91% and 92%, respectively. Class-wise analysis revealed that the Positive Predictive Value (PPV) of the model was 95%, indicating a high level of confidence in the prediction of bioactive compounds. The model was able to identify 87% of active compounds and 96% of inactive compounds, with an accuracy of 88% for inactive compounds. In conclusion, the classification report of the model highlights its exceptional ability to identify active compounds and the high accuracy of its predictions.

The initial training of the model was performed on a small dataset to demonstrate its proof of concept and establish a baseline performance. However, as the goal of the model was to accurately classify compounds in a large-scale dataset, it was deemed necessary to further train the model on a significantly larger dataset. This additional training allowed the model to improve its generalization capabilities, and to better capture the underlying patterns in the data. Moreover, a larger dataset helps the model to avoid overfitting and increase its robustness.

On the large Pubchem dataset, we trained the proposed model. This dataset, however, is extremely unbalanced. This poses an issue for the model because it may learn to make biased predictions in favor of the majority class. Synthetic Minority Over-sampling Technique (SMOTE) was used to generate synthetic samples for the minority class in order to address this issue [39]. Rather than simply duplicating existing instances, SMOTE generates synthetic samples by interpolating between existing minority class instances. This helped to balance the dataset and mitigate the class imbalance problem, resulting in a more accurate and robust model. The use of SMOTE was critical in ensuring that the model was unbiased and performed well on both minority and majority classes.

**Table 5. Performance of neural network on ChEMBL data.**

|  | Precision | Recall | f1-score | Support |
|---|---|---|---|---|
| **0 (Inactive)** | 0.88 | 0.96 | 0.92 | 23 |
| **1 (Active)** | 0.95 | 0.87 | 0.91 | 23 |
| **Accuracy** | 0.91 | | | 46 |

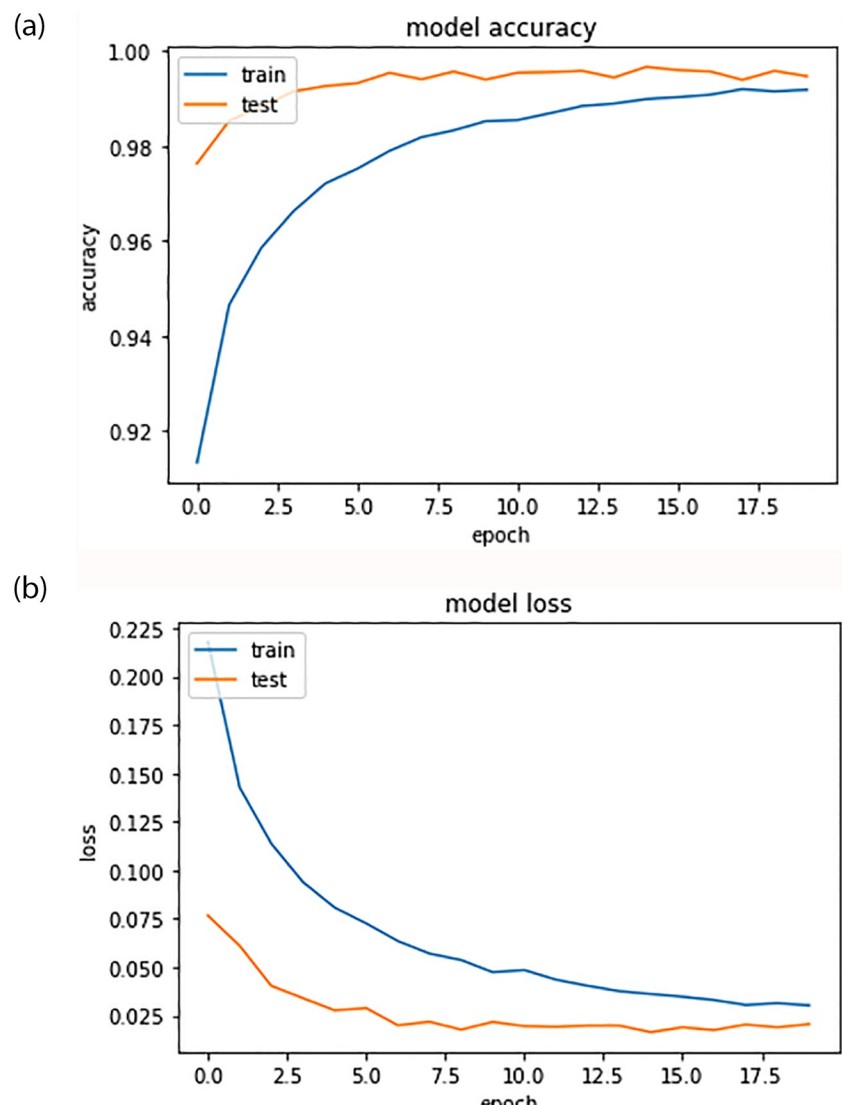

**Fig 2. Learning curve for training and validation data.**

The results of the large-scale training revealed significant improvement in the model's performance, providing evidence for the importance of training deep learning models on a large, diverse dataset for optimal results. Fig 2 depicts the learning progression of the training and validation phases. Table 6 displays the classification performance. The test set included

**Table 6. Performance of neural network on Pubchem data.**

|  | Precision | Recall | f1-score | Support |
|---|---|---|---|---|
| **0 (Inactive)** | 1.00 | 1.00 | 1.00 | 39855 |
| **1 (Active)** | 1.00 | 1.00 | 1.00 | 39854 |
| **Accuracy** | 1.00 | | | 79709 |

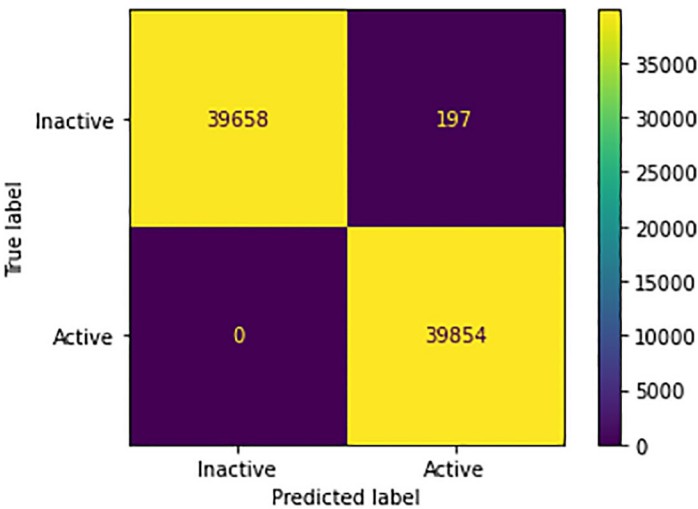

**Fig 3. Confusion matrix of neural network on pubchem data.**

approximately 80,000 samples, and the model demonstrated remarkable accuracy in class predictions. Fig 3 shows the confusion matrix for the trained model.

To demonstrate the robustness and generalization capabilities of the pre-trained model (trained from Pubchem and synthetic data), we used a combined dataset from two different data repositories (ChEMBL, Pubchem). By testing the model on a merged dataset, we can clarify its consistency and reliability, as well as its performance in a real-world scenario. The validation results on the merged dataset confirmed the effectiveness of the pre-trained model as well as its suitability for use in a variety of real-world applications. Table 7 shows the evaluation metrics and Fig 4 depicts the confusion matrix in prediction on test set. The results indicated that the pre-trained model displayed a high level of accuracy, with a 93% prediction rate for bioactivity.

## 4.3 Important fingerprints

It is important to identify specific features from PaDEL descriptors because it can lead to a better understanding of the molecular properties. These characteristics are critical in determining bioactivity, improving model performance and efficiency, and providing insights into the underlying biology and chemical mechanisms. Therefore, we have used SHAP (SHapley Additive exPlanations) on XGB classifier to find important fingerprints from the PaDEL descriptors for this task [40]. Compared to more conventional approaches like feature importance scores, this offers a more thorough and understandable representation of the feature importances. The interaction between features is taken into account by SHAP values, enabling a more in-depth knowledge of how each feature affects the model's prediction. Fig 5 demonstrates the

**Table 7. Classification performance of trained model on combined data.**

|  | Precision | Recall | f1-score | Support |
|---|---|---|---|---|
| **0 (Inactive)** | 0.88 | 1.00 | 0.94 | 751 |
| **1 (Active)** | 1.00 | 0.87 | 0.93 | 751 |
| **Accuracy** | **0.93** | | | 1502 |

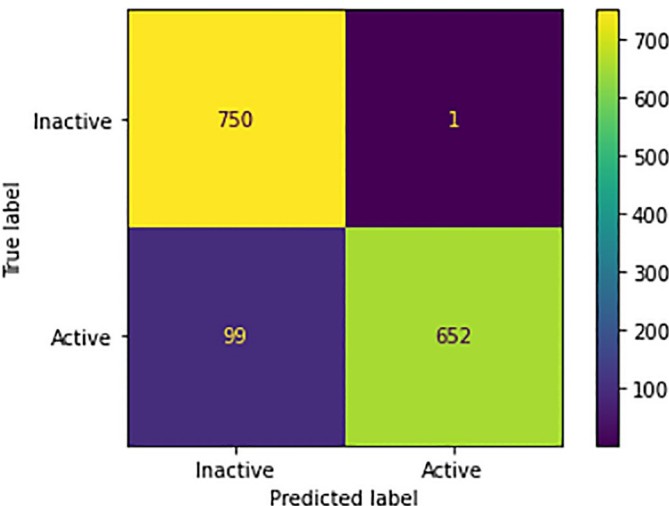

**Fig 4. Confusion matrix of pre-trained model.**

top 20 fingerprints in a molecule's PaDEL descriptors that are necessary for it to be bioactive against SARS-CoV-2. The figure shows that the bioactivity against SARS-CoV-2 has an inverse relationship with FP671, FP391, FP643, FP33, and FP368. Their widespread presence has a detrimental impact on bioactivity. However, FP341, FP20, FP180, FP143, and FP393 have a positive effect on bioactivity. Table 8 shows the corresponding substructure of these fingerprints.

## 5 Discussion

The present study utilized classification and ensemble methods on a carefully curated dataset. The results indicated that the PaDEL descriptors were effective in predicting bioactivity, and the proposed neural network design was efficient. Additionally, support vector-based classifiers, due to their advantage in handling high-dimensional data and limited number of samples, demonstrated good performance. The neural network models were able to identify the key features contributing to classification through in-depth learning, resulting in high accuracy in predicting the bioactivity class. To prevent overfitting, the neural network architecture was equipped with a dropout layer and synthetic data was used in training.

Recently, machine learning techniques have been utilized to find potential drugs against SARS-CoV-2. Besides generating effective molecular descriptors for coronaviruses, ML is also used to predict the bioactivity of existing drugs. The use of ML algorithms has been limited to in-vitro and in-vivo experiments in a few studies. A study by [26] showed that the SVM classification algorithm had an accuracy of 88%. However, our method performed better, with a 93% accuracy rate. Additionally, we investigated multiple molecular description methods and identified important molecular substructure that impacts the bioactivity.

Finding a suitable classification system to forecast the bioactivity of compounds against the SARS coronavirus was the aim of this research. We also wanted to learn the best method for describing molecules during this process. After careful curation and processing we have used experimental data from two database—ChEMBL and Pubchem, and trained a neural network model that can identify bioactivity of molecular compounds against SARS-Cov-2 with 93% accuracy. We also identified the important substructure of the molecules that impacts the bioactivity positively and negatively. However, in-vitro and in-vivo experiments can establish our findings with a probable drug structure.

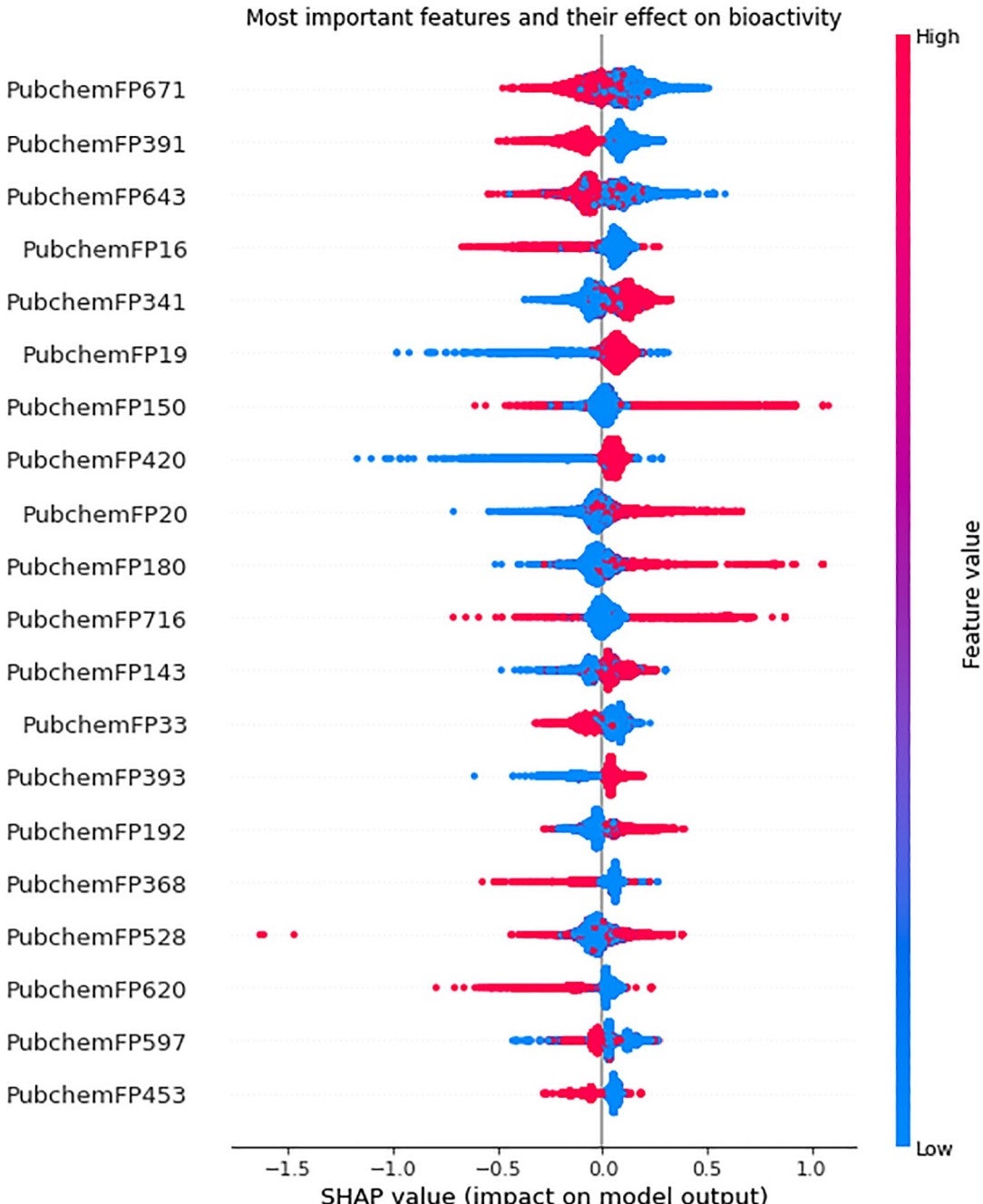

**Fig 5. Important features and relation to bioactivity.**

## 6 Finding drug candidate

We compared our results with DrugCentral REDIAL 2020 by [41], an activity estimation portal against coronavirus, to evaluate our neural network model using a list of different small molecules targeting SARS-CoV-2 3CLpro used by [28]. There are approximately 700 inactive compounds and 486 SARS-CoV-2 3CLPro inhibitors in this list. With a threshold class probability of 99.9%, our model correctly classified 456 chemicals as active. We created several

**Table 8. Important fingerprints for bioactivity prediction against SARS-CoV-2.**

| FP Bit position | Substructure | Effect on Bioactivity |
|---|---|---|
| FP671 | O = C-C = C-C | Negative |
| FP391 | N($\sim$C)($\sim$C)($\sim$C) | Negative |
| FP643 | [#1]-C-C-N-[#1] | Negative |
| FP33 | >= 1 S | Negative |
| FP368 | C($\sim$H)($\sim$S) | Negative |
| FP341 | C($\sim$C)($\sim$C)($\sim$O) | Positive |
| FP20 | >= 4 O | Positive |
| FP180 | >= 1 saturated or aromatic nitrogen-containing ring size 6 | Positive |
| FP143 | >= 1 any ring size 5 | Positive |
| FP393 | N($\sim$C)($\sim$H) | Positive |

activities of the same compound from REDIAL 2020 to further validate the active compounds we have discovered from our model, giving us a notion that these compounds can be considered as drug candidates against SARS-Cov-2 3CLPro. The activities derived from REDIAL are the following:

- SARS-CoV-2 cytopathic effect (CPE)

- SARS-CoV-2 cytopathic effect (host tox Counter) / Cytotoxicity

- ACE2 enzymatic activity

- 3CL enzymatic activity

- SARS-CoV pseudotyped particle entry (CoV-PPE)

**Table 9. Activities of the candidate compounds from REDIAL 2020, where, Active = A, Inactive = I, Cytotoxicity= $C_{yt}$, CoV-PPE=$C_p$, MERS-PPE= $M_p$, CoV--PPE_cs=$C_{pcs}$, MERS-PPE_cs= $M_{pcs}$.**

| PubChem CID | CPE | $C_{yt}$ | ACE2 | 3CL | $C_p$ | $M_p$ | $C_{pcs}$ | $M_{pcs}$ | hCYTOX |
|---|---|---|---|---|---|---|---|---|---|
| 49778034 | A | A | A | A | A | A | A | A | A |
| 59218137 | A | A | A | I | A | A | I | A | A |
| 68986845 | A | A | A | I | A | A | A | A | A |
| 57555613 | A | I | A | I | A | A | A | A | A |
| 9932218 | I | A | A | A | A | A | A | A | A |
| 58923182 | A | I | I | I | A | I | I | A | A |
| 15980574 | A | A | A | A | A | A | A | A | A |
| 44363685 | I | I | A | A | I | I | A | A | A |
| 59218103 | A | A | A | I | A | A | A | A | A |
| 68985847 | A | A | A | I | A | A | A | A | A |
| 59218156 | A | A | A | I | A | A | A | A | A |
| 59218176 | A | I | A | I | A | A | I | A | A |
| 24851823 | A | A | A | I | A | A | I | I | A |
| 68985777 | A | A | A | I | A | A | A | A | A |
| 68988817 | A | A | A | I | A | A | A | A | A |
| 68983302 | A | A | A | I | A | A | A | A | A |
| 68986364 | A | A | A | I | A | A | A | A | A |
| 11840037 | A | A | A | A | A | A | A | A | A |
| 21898262 | A | I | A | A | A | A | A | A | A |
| 68984823 | A | A | A | I | A | A | A | A | A |

- MERS-CoV pseudotyped particle entry (MERS-PPE)

- SARS-CoV pseudotyped particle entry counter screen (CoV-PPE_cs)

- MERS-CoV pseudotyped particle entry counter screen (MERS-PPE_cs)

- Human fibroblast toxicity (hCYTOX)

In Table 9, we present 20 active chemicals identified by our model together with their level of activity. It is obvious that our model's active chemicals are supported by these actions to be candidates for new drug development. All of the discovered active compounds' SMILES representations are accessible in the Github repository—https://github.com/fbabd/bioactivity-against-SARS-CoV-2.

## 7 Conclusion

It is evident that the drug development is a costly and time-consuming process. Through using the proper molecular descriptors and the capabilities of machine learning techniques this process could well be expedited efficiently. The bioactivity of a drug against the SARS-CoV-2 3CLpro protein was determined in this study using classic machine learning and ensemble approaches. The study examined the Lipinski and PaDEL molecular descriptors to determine whether one is more appropriate for such classification activity. Subsequently, this study also present an efficient neural network model which surpassed the competition with just prediction performance of 93%. Our model was trained on a large dataset that was carefully curated and collected from two different data sources. The proposed model was trained in a large dataset that has been vetted and assembled from two distinct data sources. In addition, the trained model was then applied to a list of 1186 candidate compounds of which 486 identified active inhibitors. The model identified 456 compounds as active, which were then evaluated for further activities against SARS-CoV-2 using REDIAL. Therefore, it is demonstrated in our approach can effectively determine the undiscovered active substance. To determine which molecular structure will be the greatest option for drug creation, more in-vitro and in-vivo study is necessary. In addition, the study identified a significant substructure of molecules that impacts the bioactivity against SARS-Cov-2. Biologists benefit from bioactivity prediction in drug design because it allows for in silico screening of potential drug candidates prior to conducting costly and time-consuming experiments. By reducing the number of unsuccessful compounds that must be tested, this helps to save resources and increase efficiency in the drug discovery process. This work will eliminate the laborious search for potential molecules against the deadly SARS-CoV-2 reasonably.

## Supporting information

**S1 Graphical abstract.**
(PNG)

## Author Contributions

**Conceptualization:** Sumona Hoque Mumu.

**Data curation:** Sanjida Akter, Sumona Hoque Mumu.

**Formal analysis:** Faisal Bin Ashraf, Sumona Hoque Mumu.

**Funding acquisition:** Jasim Uddin.

**Investigation:** Sanjida Akter, Sumona Hoque Mumu, Muhammad Usama Islam, Jasim Uddin.

**Methodology:** Faisal Bin Ashraf.

**Project administration:** Jasim Uddin.

**Resources:** Sanjida Akter.

**Software:** Faisal Bin Ashraf, Sumona Hoque Mumu.

**Supervision:** Muhammad Usama Islam, Jasim Uddin.

**Validation:** Muhammad Usama Islam.

**Visualization:** Faisal Bin Ashraf, Muhammad Usama Islam.

**Writing – original draft:** Faisal Bin Ashraf, Muhammad Usama Islam, Jasim Uddin.

**Writing – review & editing:** Muhammad Usama Islam, Jasim Uddin.

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
