## [Decision Letter · Decision Letter 0]

30 Mar 2023

PONE-D-23-05165Bio-activity Prediction of Drug Candidate Compounds Targeting SARS-Cov-2 using Machine Learning ApproachesPLOS ONE

Dear Dr. Uddin,

Thank you for submitting your manuscript to PLOS ONE. After careful consideration, we feel that it has merit but does not fully meet PLOS ONE’s publication criteria as it currently stands. Therefore, we invite you to submit a revised version of the manuscript that addresses the points raised during the review process.

We look forward to receiving your revised manuscript.

Kind regards,

Ahmed A. Al-Karmalawy, Ph.D.

Academic Editor

PLOS ONE

Journal Requirements:

   "NO - The funders had no role in study design, data collection and analysis, decision to publish, or preparation of the manuscript."

Reviewers' comments:

Reviewer's Responses to Questions

**Comments to the Author**

1. Is the manuscript technically sound, and do the data support the conclusions?

Reviewer #1: Yes

Reviewer #2: Yes

Reviewer #3: Yes

2. Has the statistical analysis been performed appropriately and rigorously? 

Reviewer #1: Yes

Reviewer #2: N/A

Reviewer #3: N/A

3. Have the authors made all data underlying the findings in their manuscript fully available?

Reviewer #1: Yes

Reviewer #2: Yes

Reviewer #3: Yes

4. Is the manuscript presented in an intelligible fashion and written in standard English?

Reviewer #1: Yes

Reviewer #2: Yes

Reviewer #3: Yes

5. Review Comments to the Author

Reviewer #1: The presented manuscript provides a bioactivity prediction approach for small molecules targeting SARS-CoV-2. The authors constructed a machine learning model for investigating several molecular descriptors of active compounds deposited within different databases to define the optimal descriptor for activity prediction. The proposed model was shown efficient as it successfully identified positive anti-SARS-CoV-2 drug candidates. Findings are promising and would redeem future investigation and development. The manuscript is relevant, valuable in the field, and with potentiality for high citation. Publication of the manuscript is recommended following several suggestions as they are as following:

1. The authors should provide adequate rational for adopting Lipinski versus PaDEL as comparative tools for calculating molecular descriptive. The comparison isn’t equally footed since, unlike PaDEL, Lipinski’s tool provides only physiochemical parameters and constitutional descriptors, reflecting the ligand’s chemical composition without any information about the molecular geometry nor the atom connectivity. Other molecular descriptor calculators should be used instead like Mordred or else.

Additionally, natural products are often cited as an exception to Lipinski's rules, the thing that would ignore potential candidates.

2. Authors should provide reference for the criteria used in categorizing the compound’s activity (Active, Inactive, and moderate).

3. Parameter tuning is generally crucial for machine learning performances. Authors should clearly illustrate their tuning approach.

4. Figures are provided without any legend description.

5. Author should be careful when converting reference styles to match the submitted journal guidelines. Several paragraphs started by reference numbers rather than the author name. for example, lines 92 and 105. Authors are advised to adequately provide proper in-text citation throughout the whole manuscript.

Reviewer #2: The manuscript entitled “Bio-activity Prediction of Drug Candidate Compounds Targeting SARS-Cov-2 using Machine Learning Approaches” reports evaluation of 27 machine learning classifiers. Authors have developed a neural network model that could identify bioactive and inactive chemicals. Abstract section should also provide a quick overview for the obtained results and its implications. Other than that, study methodologies, results and discussion are well presented.

Reviewer #3: manuscript titled: Bio-activity Prediction of Drug Candidate Compounds Targeting SARS-Cov-2 using

Machine Learning Approaches provide a machine learning approach to target the SARS-CoV-2 3CLpro protein, the manuscript is well written and well presented, my only comment is that the Figures need to be of higher quality.

6. PLOS authors have the option to publish the peer review history of their article (what does this mean?). If published, this will include your full peer review and any attached files.

Reviewer #1: No

Reviewer #2: No

Reviewer #3: No

---

## [Author Response · Author response to Decision Letter 0]

15 May 2023

The additional file 'Response to the Reviewers' has been attached.

---

## [Decision Letter · Decision Letter 1]

4 Jun 2023

PONE-D-23-05165R1Bio-activity Prediction of Drug Candidate Compounds Targeting SARS-Cov-2 using Machine Learning ApproachesPLOS ONE

Dear Dr. Jasim Uddin,

Thank you for submitting your manuscript to PLOS ONE. After careful consideration, we feel that it has merit but does not fully meet PLOS ONE’s publication criteria as it currently stands. Therefore, we invite you to submit a revised version of the manuscript that addresses the points raised during the review process.

We look forward to receiving your revised manuscript.

Kind regards,

Ahmed A. Al-Karmalawy, Ph.D.

Academic Editor

PLOS ONE

Journal Requirements:

Additional Editor Comments:

Reviewer 4 has suggested that you cite specific references. You are welcome to add it/them, if you believe they are relevant. However, you are not required to include these citations, and if you do not include them, this will not influence my decision.

Reviewers' comments:

Reviewer's Responses to Questions

**Comments to the Author**

1. If the authors have adequately addressed your comments raised in a previous round of review and you feel that this manuscript is now acceptable for publication, you may indicate that here to bypass the “Comments to the Author” section, enter your conflict of interest statement in the “Confidential to Editor” section, and submit your "Accept" recommendation.

Reviewer #1: All comments have been addressed

Reviewer #4: (No Response)

2. Is the manuscript technically sound, and do the data support the conclusions?

Reviewer #1: Yes

Reviewer #4: Yes

3. Has the statistical analysis been performed appropriately and rigorously? 

Reviewer #1: Yes

Reviewer #4: Yes

4. Have the authors made all data underlying the findings in their manuscript fully available?

Reviewer #1: Yes

Reviewer #4: Yes

5. Is the manuscript presented in an intelligible fashion and written in standard English?

Reviewer #1: Yes

Reviewer #4: Yes

6. Review Comments to the Author

Reviewer #1: The authors kindly responded to all raised concerns and recommendations. The can be accepted for publication.

Reviewer #4: In this study, the authors developed a method to predict the bioactivity of compounds against the SARS-CoV-2 3CLpro protein using machine learning techniques. They utilized PaDEL descriptors and a specially designed neural network model, achieving an impressive 93% accuracy rate. The authors also identified important molecular substructures impacting bioactivity. The results, while promising, require further validation through in-vitro and in-vivo experiments. The manuscript has been written regular with a good discussion.

Thus, this manuscript can be consider to be accepted after minor modification. There are some revisions for better understanding as below:

1) The authors' choice to employ the PaDEL software for generating molecular descriptors is of particular interest. Could the authors elucidate the considerations and comparisons that led them to choose PaDEL, especially when considering potential alternatives such as Gaussian, Dragon, and ChemOffice…? What advantages did PaDEL offer in this particular context that might have superseded the benefits of the aforementioned alternatives?

2) In the Lipinski Descriptors section, please add these references to enrich this part (https://doi.org/10.1371/journal.pone.0284539;
https://doi.org/10.1080/07391102.2023.2212304; )

3) The authors indicate that machine learning techniques have been applied to predict the bioactivity of existing drugs. Could they expand on this point, and possibly discuss any significant findings or implications in this area?

7. PLOS authors have the option to publish the peer review history of their article (what does this mean?). If published, this will include your full peer review and any attached files.

Reviewer #1: **Yes**

Reviewer #4: No

---

## [Author Response · Author response to Decision Letter 1]

16 Jun 2023

Response to the reviewer file has been uploaded.

---

## [Decision Letter · Decision Letter 2]

19 Jun 2023

Bio-activity Prediction of Drug Candidate Compounds Targeting SARS-Cov-2 using Machine Learning Approaches

PONE-D-23-05165R2

Dear Dr. Jasmin Uddin,

We’re pleased to inform you that your manuscript has been judged scientifically suitable for publication and will be formally accepted for publication once it meets all outstanding technical requirements.

Kind regards,

Ahmed A. Al-Karmalawy, Ph.D.

Academic Editor

PLOS ONE

Reviewers' comments:

Reviewer's Responses to Questions

**Comments to the Author**

1. If the authors have adequately addressed your comments raised in a previous round of review and you feel that this manuscript is now acceptable for publication, you may indicate that here to bypass the “Comments to the Author” section, enter your conflict of interest statement in the “Confidential to Editor” section, and submit your "Accept" recommendation.

Reviewer #4: All comments have been addressed

2. Is the manuscript technically sound, and do the data support the conclusions?

Reviewer #4: Yes

3. Has the statistical analysis been performed appropriately and rigorously? 

Reviewer #4: Yes

4. Have the authors made all data underlying the findings in their manuscript fully available?

Reviewer #4: Yes

5. Is the manuscript presented in an intelligible fashion and written in standard English?

Reviewer #4: Yes

6. Review Comments to the Author

Reviewer #4: Based on the provided information, it seems that the authors have effectively addressed all the concerns and recommendations raised during the review process. As a result, it is recommended that the thesis be accepted for publication.

7. PLOS authors have the option to publish the peer review history of their article (what does this mean?). If published, this will include your full peer review and any attached files.

Reviewer #4: No

---

## [Editor Report · Acceptance letter]

6 Jul 2023

PONE-D-23-05165R2 

Bio-activity Prediction of Drug Candidate Compounds Targeting SARS-Cov-2 using Machine Learning Approaches 

Dear Dr. Uddin:

I'm pleased to inform you that your manuscript has been deemed suitable for publication in PLOS ONE. Congratulations! Your manuscript is now with our production department. 

Kind regards, 

on behalf of

Dr. Ahmed A. Al-Karmalawy 

Academic Editor

PLOS ONE